# Olefin-accelerated solid-state C–N cross-coupling reactions using mechanochemistry

Koji Kubota[1], Tamae Seo[1], Katsumasa Koide[1], Yasuchika Hasegawa[1,2] & Hajime Ito [1,2]

Palladium-catalyzed cross-coupling reactions are one of the most powerful and versatile methods to synthesize a wide range of complex functionalized molecules. However, the development of solid-state cross-coupling reactions remains extremely limited. Here, we report a rational strategy that provides a general entry to palladium-catalyzed Buchwald-Hartwig cross-coupling reactions in the solid state. The key finding of this study is that olefin additives can act as efficient molecular dispersants for the palladium-based catalyst in solid-state media to facilitate the challenging solid-state cross-coupling. Beyond the immediate utility of this protocol, our strategy could inspire the development of industrially attractive solvent-free palladium-catalyzed cross-coupling processes for other valuable synthetic targets.

[1] Division of Applied Chemistry, Graduate School of Engineering, Hokkaido University, Sapporo, Hokkaido 060-8628, Japan. [2] Institute for Chemical Reaction Design and Discovery (WPI-ICReDD), Hokkaido University, Sapporo, Hokkaido 060-8628, Japan. Correspondence and requests for materials should be addressed to K.K. (email: kbt@eng.hokudai.ac.jp) or to H.I. (email: hajito@eng.hokudai.ac.jp)

Historically, most organic transformations have been carried out in solution. Such solution-based organic syntheses generally require liquid organic solvents to dissolve reactants or catalysts in a reaction flask. Accordingly, the pharmaceutical industry and the fine chemicals industry strongly depend on solvent-based organic synthesis, which has led to serious problems with regard to solvent waste, as organic solvents usually account for ~80–90% of the total mass used in any organic reaction[1–3]. Although solvent recycling is a very effective way to reduce solvent waste, organic chemists should focus on (re)designing organic syntheses to use less or no solvent. In this context, solid-state organic transformations have attracted considerable attention as cleaner and sustainable synthetic alternatives[4–7]. In addition, these methods would be exciting opportunities to access large areas of hitherto unexplored chemical space that exhibit different reactivity and selectivity compared to conventional solution-based reactions[8–20].

Palladium-catalyzed cross-coupling reactions have long been used as arguably the most powerful, versatile, and well-established organic transformations with broad applications ranging from natural product synthesis and medicinal chemistry to polymer and materials science[21–24]. Despite recent significant progress, the exploration of new strategies, reaction media, and concepts for the improvement of the sustainability of cross-coupling reactions still remains an important and challenging research subject. Conventionally, palladium-catalyzed cross-coupling reactions of liquid and solid substrates are conducted in organic solvents (Fig. 1a)[21,22]. When using liquid substrates, the cross-coupling reactions can in some cases be carried out under neat conditions, where liquid substrates serve as reactants and reaction solvent (Fig. 1a)[25–32]. Whereas the benefits of these solution-based reactions are well-established, the value of palladium-catalyzed cross-coupling processes becomes even more apparent when solid-state cross-coupling reactions are considered, especially in the context of solvent-waste prevention and environmental protection[4–7]. In addition, solid-state coupling reactions should be particularly useful for poorly soluble substrate classes such as large polycyclic aromatics due to strong intermolecular interactions (e.g., π–π interactions). Nevertheless, the solid state has remained extremely limited as a reaction medium for palladium-catalyzed cross-coupling processes (Fig. 1a)[21,22]. Thus, we sought to re-design palladium-based catalyst systems for the solid state, which could potentially unlock versatile applications for solid-state synthesis.

Mechanochemical solvent-free reactions using ball milling or milling with a catalytic amount of liquid, the so-called liquid-assisted grinding (LAG), have emerged as powerful alternatives to synthesis in solution[33–40], and mechanochemical palladium-catalyzed cross-coupling reactions have already been reported[41–53]. However, these methods focus mostly on neat liquids[41–46,48–53]. For solid-state substrates, the scope is significantly more restricted and low conversion rates are common[42,47,49]. Here, we report the development of a potentially general and scalable solvent-free method for solid-state palladium-catalyzed C–N cross-coupling reactions using mechanochemistry (Fig. 1b)[54]. The key finding in this study is that the addition of a small amount of olefin dramatically accelerates these challenging solid-state cross-coupling reactions. Based on a transmission electron microscopy (TEM) analysis, we discovered that some such olefin additives can act as efficient molecular dispersants for palladium catalysts in solid-state media to inhibit undesired aggregation of the catalyst that may lead to catalyst deactivation (Fig. 1c). The protocol should be particularly useful for the rapid access to structurally complex triarylamines, which can be found in a wide range of organic materials including solar cells and light-emitting diodes (Fig. 1b)[55–57]. In fact, we will demonstrate (vide infra) that when using the solid-state cross-coupling reaction presented herein, arylamine-based hole-transporting materials can be prepared faster and in better yield relative to conventional methods[57]. Thus, we anticipate that the present solvent-free solid-state palladium-catalyzed cross-coupling reactions may potentially find broad applications in industrially relevant syntheses.

## Results

**Development of solid-state C–N cross-coupling.** All reactions were conducted in a Retsch MM400 mill (stainless-steel milling jar; 30 Hz; stainless-steel balls). Initially, we compared the reactivity of liquid 1-bromonaphthalene (**1a**) and solid 1-bromopyrene (**1b**) in the palladium-catalyzed C–N cross-coupling reaction with diphenylamine **2a** under solvent-free mechanochemical conditions (Fig. 2). Very recently, Su and co-workers have reported the mechanochemical palladium-catalyzed C–N cross-coupling of aryl chloride using NaCl as a mechanochemical auxiliary[51]. Even though this development is indisputably remarkable, the substrate scope mostly focused on liquid substrates. Thus, we anticipated that the reaction of liquid substrate **1a** should proceed readily using the Pd(OAc)$_2$/XPhos (**P1**) catalyst system developed by Su[51]. Indeed, the corresponding coupling product (**3a**) was obtained in moderate yield (41% yield; Fig. 2). When we used the Pd(OAc)$_2$/t-Bu$_3$P (**P2**) catalyst system, which is a high-performance catalyst for C–N coupling that has been reported by Hartwig and co-workers[58], **3a** was obtained quantitatively (95% yield; Fig. 2). Other liquid aryl halides could also be coupled with diphenylamine **2a** in the presence of Pd(OAc)$_2$/t-Bu$_3$P (**P2**) in high yield (see Supplementary Figure 3). Then, we proceeded to investigate the C–N cross-coupling behavior of solid 1-bromopyrene (**1b**) (Fig. 2). We found that the solid-state cross-coupling reactions involving **1b** were sluggish using either the Pd(OAc)$_2$/XPhos (**P1**) or the Pd(OAc)$_2$/t-Bu$_3$P (**P2**) catalyst systems (3% and 28% yield, respectively; Fig. 2). These results suggest the presence of a considerable reactivity gap between liquid and solid substrates, even under mechanochemical conditions.

We therefore decided to focus on commonly used phosphine ligands in solvent-based systems in order to potentially facilitate the solid-state C–N cross-coupling under mechanochemical conditions (entries 1–10, Fig. 3a)[54]. Experiments involving catalyst systems consisting of 5 mol% Pd(OAc)$_2$ and 5 mol% phosphine ligand revealed that the use of bulky and highly electron-donating monophosphines such as Ad$_3$P (**P3**) provides the desired coupling product in low yield (18% yield; entry 1). In contrast, the use of Cy$_3$P (**P4**) did not afford the targeted coupling product (entry 2). The reaction did not proceed when commonly used Buchwald-type ligands such as **P5**–**P9** were employed (entries 3–7)[54]. While **P5** is a highly effective ligand for C–N cross-coupling reactions in solvents at room temperature, the desired product was not formed under solid-state conditions (entry 3)[59]. Diphosphine ligands such as rac-Binap (**P10**) and Xantphos (**P11**) were also examined, but a reaction was not observed (entries 8 and 9). Increasing the catalyst loading did not improve the product yield when **P2** was used as a ligand (33% yield; entry 10). Next, we attempted LAG, which uses substoichiometric liquid additives, to improve the reactivity (entries 11–20)[33–40]. Unless otherwise noted, the following reactions with liquid additives are all characterized by a 0.20 ratio of μL of liquid olefin added per mg of reactant. Although small amounts of toluene, benzene, and tetrahydrofuran (THF), which are commonly used organic solvents in a palladium-catalyzed C–N cross-coupling reactions[54], slightly improved the yield of **3b**, the yields remained moderate (20–55% yield; entries 11–15). Other commonly used solvents such as dimethylsulfoxide (DMSO), acetonitrile (MeCN), and hexane did not or poorly

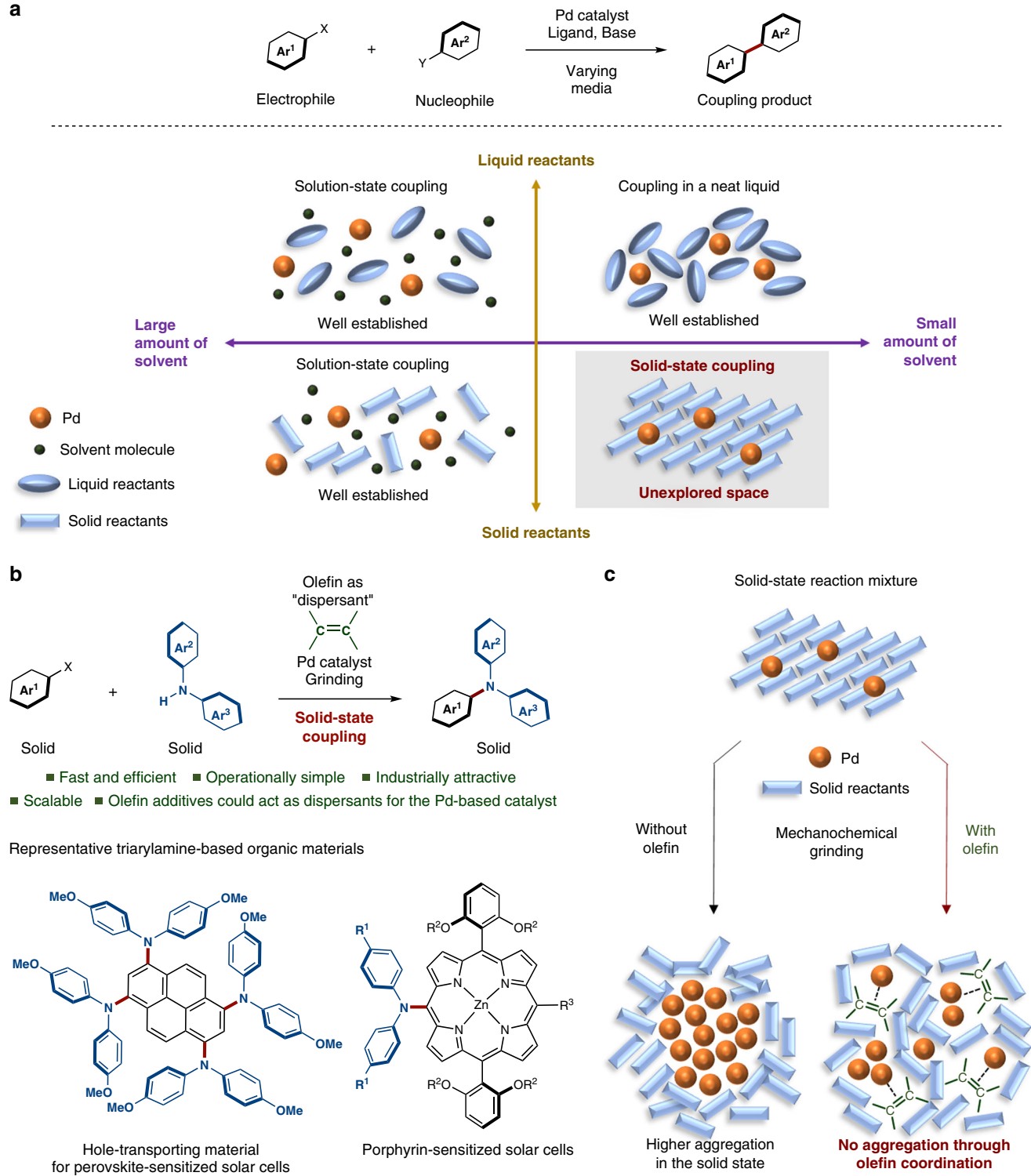

**Fig. 1** Overview of the olefin-accelerated solid-state couplings using mechanochemistry. **a** Current application range of palladium-catalyzed cross-coupling reactions. **b** General and scalable solid-state C–N cross-coupling reactions using olefin additives as molecular dispersants. **c** A proposed acceleration mechanism, wherein olefin additives could act as dispersants for catalysts in solid-state media and facilitate the solid-state cross-coupling

promote the solid-state cross-coupling (0%, 10% and 16% yield, respectively; entries 16–18). Lastly, we attempted cycloalkanes such as cyclohexane and cyclooctane, which resulted in moderate yields (54% and 46% yield, respectively; entries 19 and 20).

We speculated that one possible reason for the observed reactivity difference between liquid and solid substrates may be facile aggregation of the palladium catalysts in the solid-state

reaction mixtures that may lead to catalyst deactivation. Thus, we wondered whether olefin additives could be used as molecular dispersants for the palladium catalyst, i.e., the olefins could coordinate toward any off-cycle palladium species and suppress higher aggregation of catalysts in the solid-state medium (Fig. 3b)[60]. The following reactions with olefins are all characterized by a 0.20 ratio of microliters of liquid olefin added

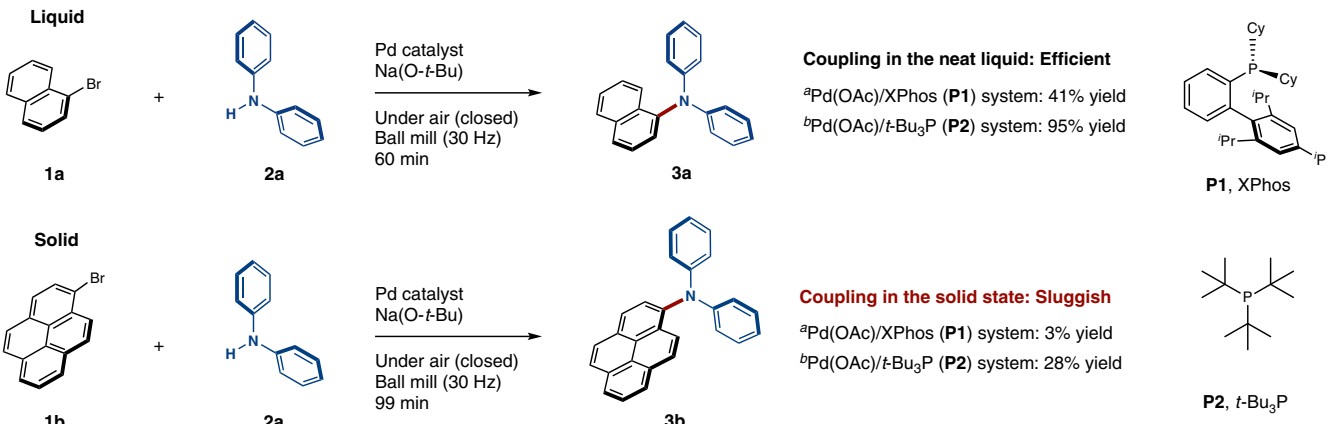

**Fig. 2** Comparison of the reactivity of liquid and solid aryl bromides. [a]The following reaction conditions[51] were used: 0.6 mmol of **1**; 0.5 mmol of **2a**; 0.01 mmol of Pd(OAc)$_2$; 0.02 mmol of XPhos (**P1**); 1.0 mmol of Na(O-t-Bu); 2.0 g of NaCl; in a stainless-steel ball-milling jar (25 mL) with two stainless-steel balls (15 mm); 30 Hz; 99 min. [b]The following reaction conditions were used: 0.5 mmol of **1**; 0.5 mmol of **2a**; 0.025 mmol of Pd(OAc)$_2$; 0.025 mmol of ligand; 0.75 mmol of Na(O-t-Bu); in a stainless-steel ball-milling jar (1.5 mL) with a stainless-steel ball (3 mm); 30 Hz; 99 min

per milligram of reactant. Initially, 1,5-cyclooctadiene (1,5-cod) was tested, given that 1,5-cod is frequently employed as a weak coordination ligand for low-valent metal complexes. Pleasingly, we found that the reaction in the presence of Pd(OAc)$_2$/t-Bu$_3$P was dramatically accelerated to form the corresponding coupling product (**3b**) in 99% yield. The use of cyclooctene as an additive also effectively promoted the reaction (96% yield). In sharp contrast, the reaction with cyclooctane did not dramatically affect the reactivity (46% yield), suggesting that the presence of an olefin functional group should be important for the observed acceleration. We noted that the colors of the solid-state reaction mixture after the reaction changed depending on the conditions applied (Fig. 4). For example, the reaction mixtures containing 1,5-cod or cyclooctane appeared as dark green waxy solids, while the reaction mixture without additives appeared as a light-yellow solid. Other olefins such as 1-hexene and (E)-hex-3-ene could also be used to facilitate the solid-state cross-coupling reaction (98% and 89% yield, respectively). In contrast, the use of hexane did not influence on the reactivity (16% yield). Cyclohexene also provided a better yield (92% yield) than cyclohexane (54% yield). Interestingly, the use of norbornadiene decreased the catalytic activity of this reaction (12% yield). This might be due to the irreversible coordination of norbornadiene to active Pd(0) species. Alkynes such as oct-4-yne also showed a moderate acceleration effect (66% yield). It should furthermore be noted that the amount of olefin used in these reactions is comparable to that of the reactant (0.20 μL mg$^{-1}$), which stands in sharp contrast to conventional solution-based reactions, where often a 10- to 100-fold excess of bulk solvent is used.

**Transmission electron microscopy**. To gain mechanistic insight into the observed acceleration effect upon adding olefins, we used transmission electron microscopy (TEM) to characterize the palladium nanoparticles generated in situ in the crude reaction mixture of **1b** with **2a** (Fig. 5). The observed image clearly shows the formation of palladium nanoparticles (approximate size: 3–5 nm) in the reaction mixture in the presence of 1,5-cod (Fig. 5a). Notably, higher aggregation of the palladium particles was not observed. On the other hand, the image obtained for the palladium species derived from the reaction mixture in the presence of cyclooctane (Fig. 5b) and in the absence of any additives (Fig. 5c) showed that the palladium species significantly aggregate into dense particles (Fig 5b, c). Although further mechanistic investigations on how the catalyst diffuses in the solid-state media

should provide a deep understanding on the reaction mechanism, such a study would be extremely challenging, i.e., the mechanochemical reaction setup is unlike to be compatible with an in situ extended X-ray absorption fine structure (EXAFS) analysis. Based on the TEM analysis, we would like to propose two possible roles for the olefin additives in these solid-state cross-coupling reactions: (1) olefins could act as dispersants for the palladium catalysts to suppress higher aggregation of the nanoparticles that may lead to catalyst deactivation;[60] (2) the leaching rates for active monomeric Pd(0) specie from the palladium nanoparticles would be accelerated by coordination from the olefins, and the dissociated Pd(0) species could be coordinated by t-Bu$_3$P and release the olefin ligands to form [(t-Bu$_3$P)Pd(0)], which could subsequently activate the C–X bond of aryl halides[61–63].

**Substrate scope of the solid-state C–N cross-coupling reaction**. To explore the scope of the present solid-state coupling reaction, various amine nucleophiles were tested (Fig. 6). Both bis(4-methylphenyl) amine (**2c**) and bis(4-methoxyphenyl)amine (**2d**) were coupled in high yield (88% and 68%, respectively) under the optimized reaction conditions. Diarylamines containing a naphthyl group (**2e** and **2f**) are also compatible with the applied solid-state conditions (72% and 81%, respectively). Conversely, the reaction did not proceed for carbazole (**2g**). As solvent-free solid-state reactions can be regarded as reactions that proceed under extremely-high-concentration conditions, the reagents and catalysts in the solid-state interact much more strongly with each other than those in solution. This could tentatively explain the observed low reactivity of **2g**, which could potentially coordinate strongly to any off-cycle palladium species, which would lead to catalyst deactivation.

Subsequently, we turned our attention to the scope of the aryl halides (Fig. 6). This reaction is characterized by a broad substrate scope and permits constructing a wide range of functionalized triarylamines containing large polycyclic hydrocarbon cores. For example, pyrene derivatives (**1b**, **1h**, **1i**) were cleanly coupled with diphenylamine nucleophile **2a** to provide **3b**, **3h**, and **3i** in good to excellent yield (93%, 94%, and 70%, respectively). Other aromatic cores such as naphthalene (**1j** and **1n**), phenanthrene (**1k**), anthracene (**1l** and **1m**), biphenyl (**1o**), terphenyl (**1p**), acenaphthene (**1q** and **1r**), triphenylene (**1s**), and fluorene (**1t**) efficiently formed the corresponding triarylamines in 71–99%. Notably, **3j** was also obtained from the corresponding 2-chloronaphthalene (73%). Triarylamines containing stilbene

**a**

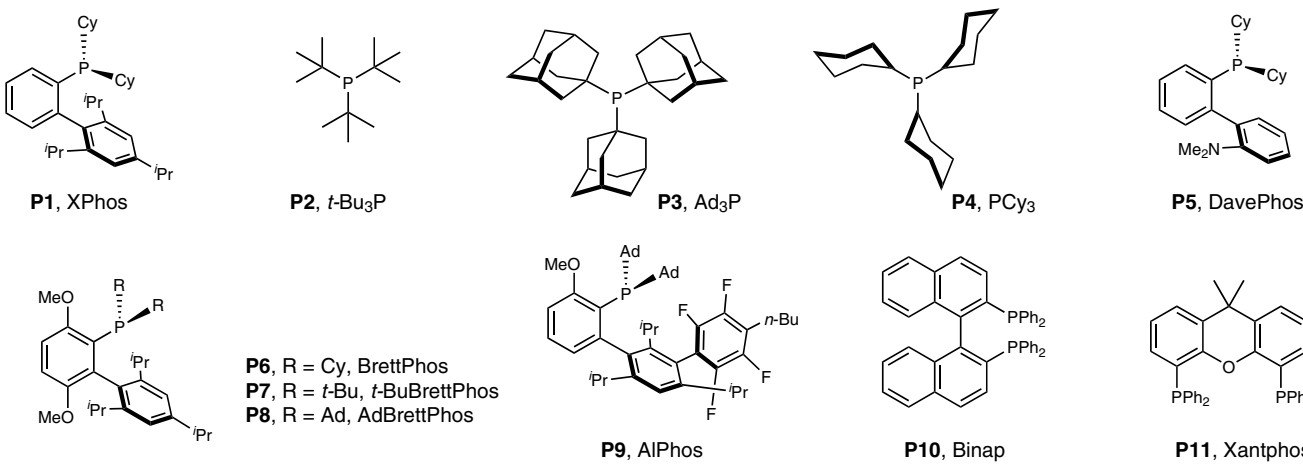

| Entry | Ligand | LAG additives | Yield (%) | | Entry | Ligand | LAG additives | Yield (%) |
|-------|--------|---------------|-----------|---|-------|--------|---------------|-----------|
| 1 | Ad₃P (**P3**) | None | 18 | | 11 | t-Bu₃P (**P2**) | Toluene | 48 |
| 2 | Cy₃P (**P4**) | None | 0 | | 12[a] | t-Bu₃P (**P2**) | Toluene | 45 |
| 3 | DavePhos (**P5**) | None | 0 | | 13[b] | t-Bu₃P (**P2**) | Toluene | 37 |
| 4 | BrettPhos (**P6**) | None | 0 | | 14 | t-Bu₃P (**P2**) | Benzene | 20 |
| 5 | t-BuBrettPhos (**P7**) | None | 0 | | 15 | t-Bu₃P (**P2**) | THF | 55 |
| 6 | AdBrettPhos (**P8**) | None | 0 | | 16 | t-Bu₃P (**P2**) | MeCN | 10 |
| 7 | AlPhos (**P9**) | None | 0 | | 17 | t-Bu₃P (**P2**) | DMSO | 0 |
| 8 | rac-Binap (**P10**) | None | 0 | | 18 | t-Bu₃P (**P2**) | Hexane | 16 |
| 9 | Xantphos (**P11**) | None | 0 | | 19 | t-Bu₃P (**P2**) | Cyclohexane | 54 |
| 10[a] | t-Bu₃P (**P2**) | None | 33 | | 20 | t-Bu₃P (**P2**) | Cyclooctane | 46 |

**P1**, XPhos   **P2**, t-Bu₃P   **P3**, Ad₃P   **P4**, PCy₃   **P5**, DavePhos

**P6**, R = Cy, BrettPhos
**P7**, R = t-Bu, t-BuBrettPhos
**P8**, R = Ad, AdBrettPhos

**P9**, AlPhos   **P10**, Binap   **P11**, Xantphos

**b**

**Olefin additives as molecular dispersants for palladium catalysts**

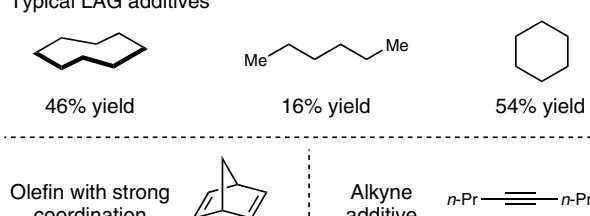

99% yield   96% yield

98% yield   89% yield   92% yield

Typical LAG additives

46% yield   16% yield   54% yield

Olefin with strong coordination   12% yield

Alkyne additive   n-Pr ═══ n-Pr   66% yield

**Fig. 3** Development of olefin-accelerated solid-state C–N cross-coupling reactions. **a** Comparison of phosphine ligands and LAG additives in solid-state C–N cross-coupling reactions. **b** Discovery of olefins as molecular dispersants for palladium catalysts. Unless otherwise noted, the following reaction conditions were used: 0.5 mmol of **1b**; 0.5 mmol of **2a**; 0.025 mmol of Pd(OAc)₂; 0.025 mmol of ligand; 0.75 mmol of Na(O-t-Bu); additive (0.20 μL mg⁻¹); in a stainless-steel ball-milling jar (1.5 mL) with a stainless-steel ball (3 mm); 30 Hz; 99 min. Yields were determined by ¹H NMR analysis using an internal standard. [a]10 mol% Pd(OAc)₂ and t-Bu₃P (**P2**) were used. [b]Toluene (0.13 μL mg⁻¹) was used

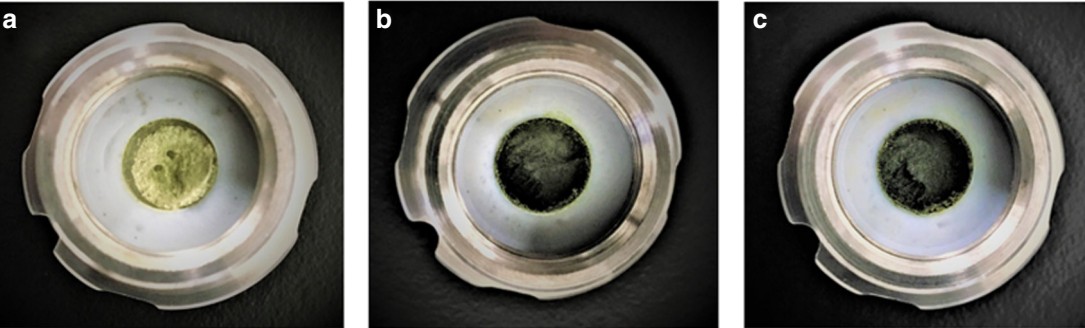

**Fig. 4** Reaction mixtures after grinding in a ball mill. Aggregation on the milling ball **a** after 99 min without additive, **b** after 99 min with 1,5-cod, and **c** after 99 min with cyclooctane

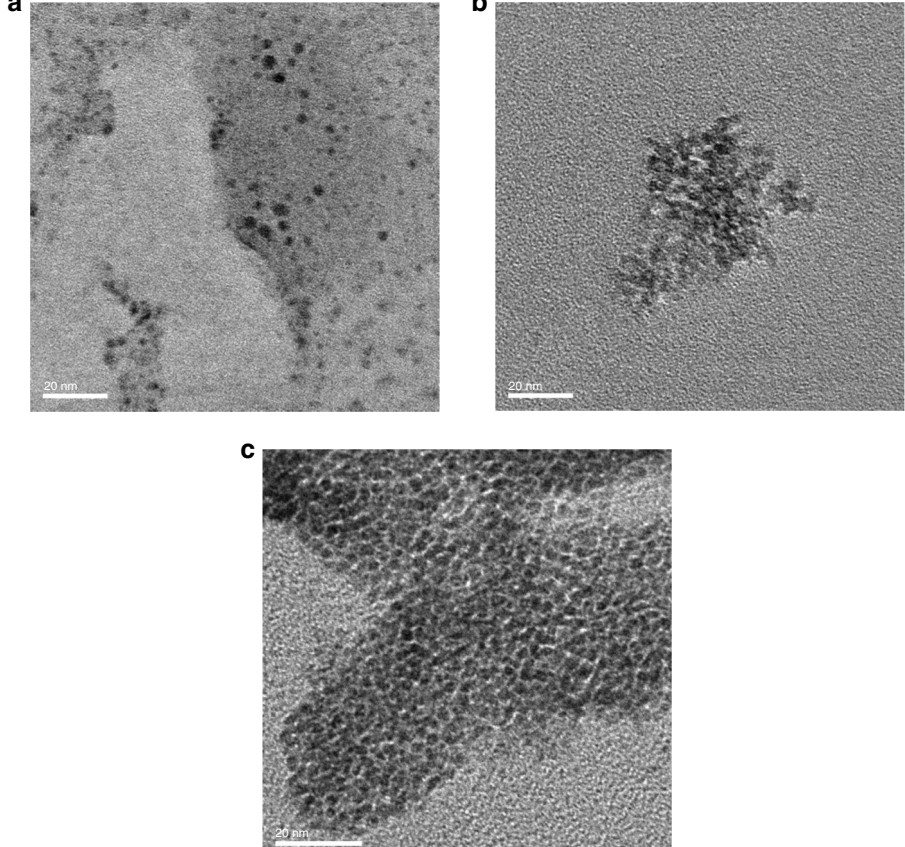

**Fig. 5** TEM images of palladium nanoparticles in the crude reaction mixtures. **a** Crude mixture after 99 min with 1,5-cod, **b** crude mixture after 99 min with cyclooctane, and **c** crude mixture after 99 min without additive. Scale bars in the TEM images (bottom left): 20 nm. These results clearly show that 1,5-cod can act as a molecular dispersant for the palladium catalyst in the solid-state reaction mixture, thus facilitating the solid-state C–N cross-coupling reaction

(**3u**) and internal alkyne (**3v**) moieties were generated in high yield (95% and 92%) under the optimized solvent-free conditions. Double aminations also proceeded smoothly to give the corresponding product (**3w**) in 91%. The reaction of **1x** afforded triarylamine-containing tetraphenylethylene **3x** (67%), which is a mechanochromic luminescent material[64]. The present strategy can also be applied to large aromatic substrates (**1y** and **1z**; 75% and 41%, respectively), which are usually poorly soluble. Finally, this method was applied to heteroatom-substituted aromatic substrates (**3aa–3ae**). Aminoporphyrins, for example, are promising and important core structures in organic materials, especially in the context of porphyrin-sensitized solar cells[55,56]. Even though palladium-catalyzed cross-coupling reactions between diarylamine nucleophiles and porphyrin electrophiles are widely used to access aminoporphyrin derivatives, such reactions commonly require considerable amounts of organic solvents, which often hampers carrying out this process in large-scale syntheses[55,56]. Pleasingly, the reaction of bromo-substituted porphyrin **1aa** proceeded efficiently under the solid-state conditions to provide aminoporphyrin **3aa** in good yield (55%). Dimesitylboryl-containing triarylamine **3ab**, which is widely known as a donor-acceptor-type charge-transfer luminescent material[65,66], can also be prepared via this solid-state cross-coupling reaction (62%). Carbazole-, thiophene-, and 1,4-benzoquinone-containing triarylamines (**3ac–3ae**) were also obtained in good yield (82%, 55%, and 57%, respectively).

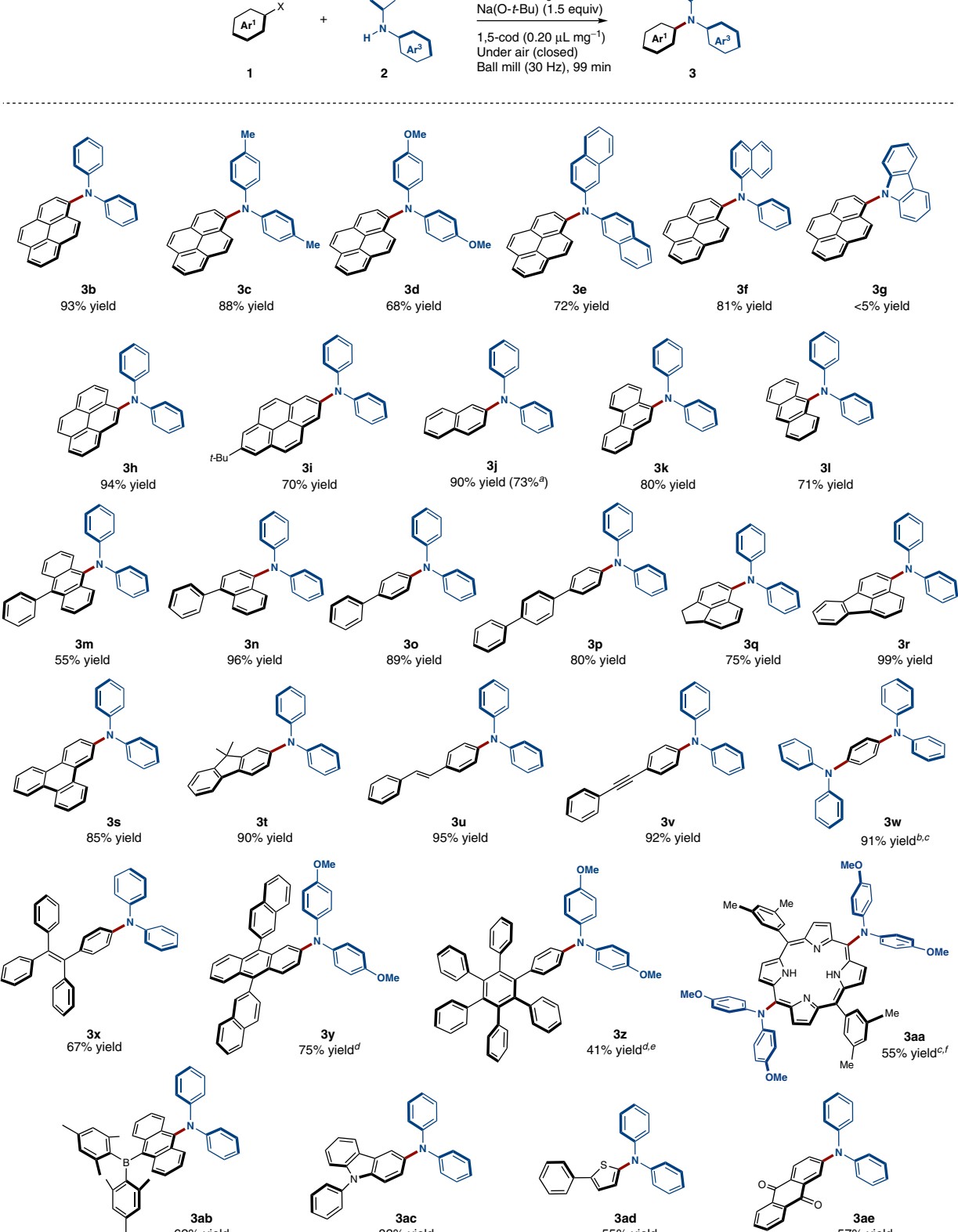

**Fig. 6** Substrate scope. Unless otherwise noted, the following reaction conditions were used: 0.5 mmol of **1**; 0.5 mmol of **2**; 0.025 mmol of Pd(OAc)$_2$; 0.025 mmol of $t$-Bu$_3$P; 0.75 mmol of Na(O-$t$-Bu); 1,5-cod (0.20 μL mg$^{-1}$); stainless-steel ball-milling jar (1.5 mL) with a stainless-steel ball (3 mm); 30 Hz; 99 min. Isolated yields are shown. [a]The aryl chloride was used as a substrate. [b]0.3 mmol scale. [c]10 mol% of catalyst and 3.0 equiv of Na(O-$t$-Bu) were used. [d]10 mol% of catalyst was used. [e]A larger stainless-steel ball-milling jar (25 mL) was used with four stainless-steel balls (10 mm). [f]0.2 mmol scale

**Solid-state coupling reactions on the gram-scale**. To demonstrate the practical utility of this protocol, we investigated the gram-scale synthesis of triarylamines under solvent-free mechanochemical conditions (Fig. 7a). The solid-state cross-coupling of **1b** with **2c** was carried out on a 7.0 mmol scale using 2 mol% palladium catalyst in a stainless-steel ball-milling jar (25 mL) with four stainless-steel balls (diameter: 10 mm), which afforded **3c** in excellent yield (92%). The product can be isolated by simple re-precipitation from CH₂Cl₂/MeOH. This result clearly demonstrates the potential utility of the present solvent-free protocol for large-scale preparations.

**Synthesis of hole-transporting materials**. Triarylamine derivatives have been extensively studied as potential organic materials for perovskite-based solar cells[55–57]. Recently, Seok and co-workers have reported that arylamine derivatives with pyrene cores could potentially be used as high-performance hole-transporting materials for perovskite-based solar cells[57]. These authors obtained tetra-substituted pyrene **3af** in moderate yield (60%)

from a solution-based Buchwald–Hartwig amination between **1af** and **2d** in toluene that was carried out at high temperature (110 °C) and required prolonged reaction times (2 days) (Fig. 7b, B). Our developed solid-state conditions, in contrast, afford **3af** in better yield (89%) after a significantly reduced reaction time (99 min) in the absence of potentially harmful organic solvents (Fig. 7b, A). This reaction illustrates that the present solid-state coupling protocol could potentially be a powerful alternative to solution-based synthetic routes to materials-science-oriented nitrogen-containing polyaromatic compounds.

**Monitoring the reaction progress**. The reaction progress of the cross-coupling reaction between **1b** and **2a** in the presence of 1,5-cod was monitored by powder X-ray diffraction (PXRD) analysis (Fig. 8). After 20 min, new diffraction peaks derived from coupling product **3b** and NaBr appeared, while the peaks associated with the starting materials remained. After 60 min, the diffraction peaks derived from the starting materials were completely disappeared, and only those of coupling product **3b** and NaBr were

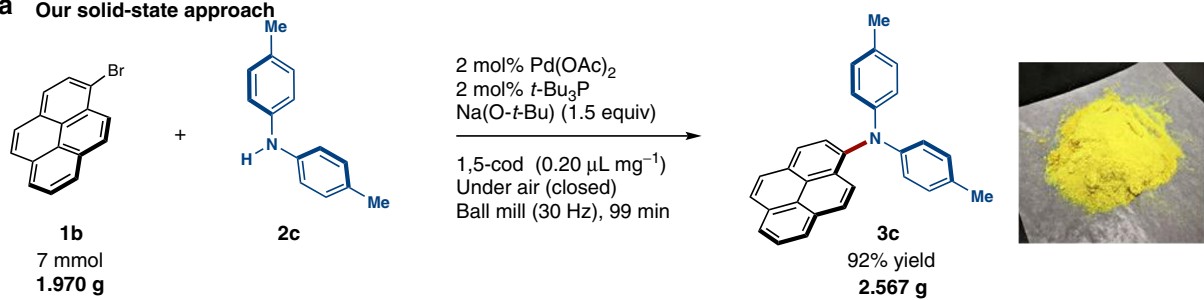

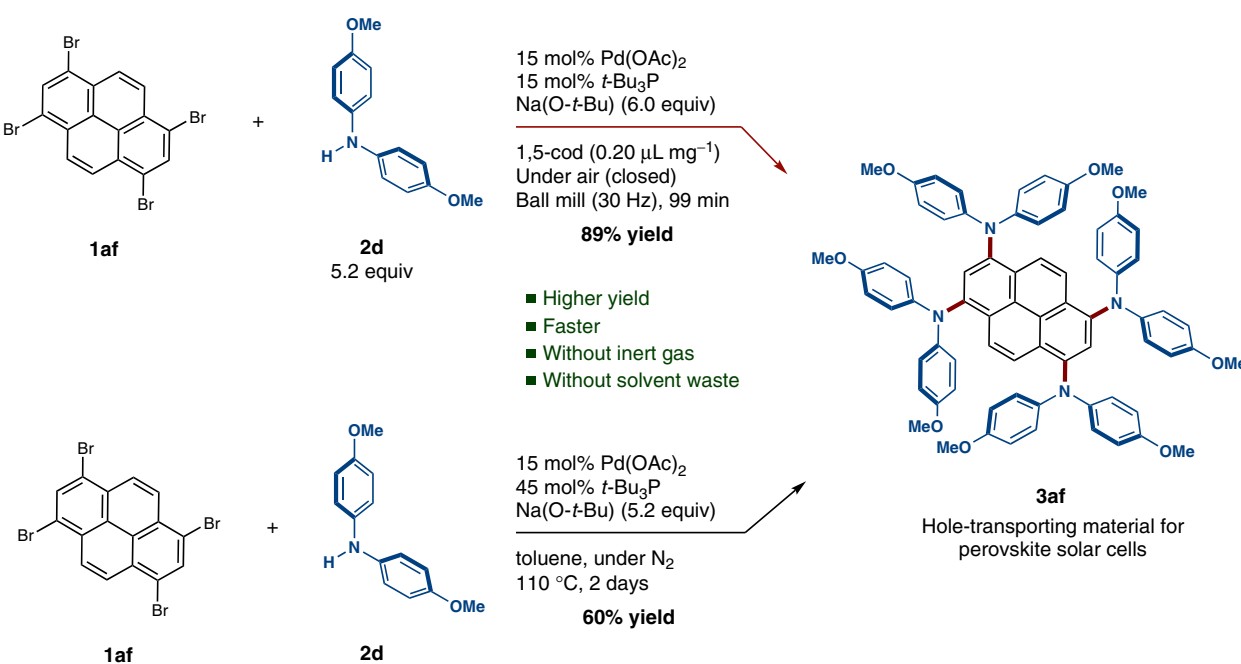

**Fig. 7** Synthetic utility of the solid-state C–N cross-coupling. **a** Solid-state gram-scale synthesis of **3c**. The following conditions were used: 7.0 mmol of **1b**; 7.0 mmol of **2c**; 0.14 mmol of Pd(OAc)₂; 0.14 mmol of t-Bu₃P; 10.5 mmol of Na(O-t-Bu); 1,5-cod (0.20 μL mg⁻¹); stainless-steel ball-milling jar (25 mL) with four stainless-steel balls (10 mm); 30 Hz; 99 min. Isolated yield is shown. **b** Efficient solid-state synthesis of the arylamine-based hole-transporting material **3af**. The following conditions were used: 0.5 mmol of **1af**; 2.6 mmol of **2d**; 0.075 mmol of Pd(OAc)₂; 0.075 mmol of t-Bu₃P; 3.0 mmol of Na(O-t-Bu); 1,5-cod (0.20 μL mg⁻¹); stainless-steel ball-milling jar (25 mL) with four stainless-steel balls (10 mm); 30 Hz; 99 min. Isolated yield is shown

observed, demonstrating a clean solid-to-solid conversion without melting during this transformation. It should also be noted that the temperature inside the milling jar after the grinding at 30 Hz for 60 min was about 30 °C, which was confirmed by thermography, indicating that this reaction proceeded at around room temperature (Supplementary Figure 6). The possibility to monitor the progress of this solid-state palladium-catalyzed synthesis in situ by PXRD is another potential advantage, given that the generation of inorganic salts can be detected, which might hold key information to understanding the underlying reaction mechanism.

**Kinetic study.** Recently, James and co-workers discovered unusual sigmoidal kinetics in the mechanochemical Knoevenagel condensation of vanillin and barbituric acid, in which the

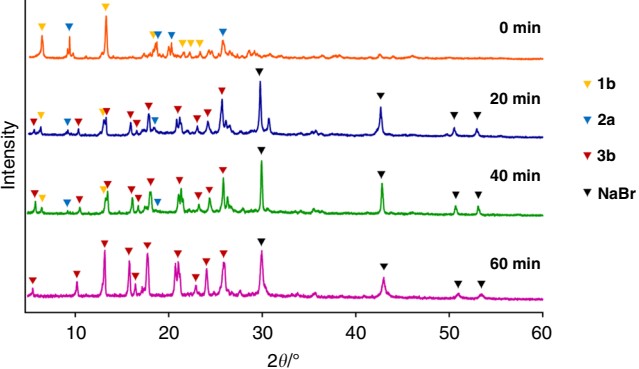

**Fig. 8** Monitoring the reaction progress by PXRD analysis. After 60 min, the diffraction peaks derived from the starting materials completely disappear, while those associated with coupling product **3b** and NaBr emerge, which suggests a clean solid-to-solid conversion without melting during the reaction

physical form of the reaction mixture dramatically changes from a free-flowing powder to a plastic-like appearance during the reaction[67]. Based on the several control experiments, they conclude that this dramatical change in rheology results in a rapid increase in the reaction rate. To address whether the rate increase is due to the olefin or the change in rheology, the kinetics of the reactions under the different conditions were measured (Fig. 9a). As periodic sampling of the reaction runs requires stopping the mill and opening the jar, each data point was obtained from an individual reaction. The kinetics of the reaction between **1b** and **2a** in the presence of 1,5-cod were found to be relatively straightforward, i.e., they could be satisfactorily modeled by simple first-order kinetics (Fig. 9a). This result suggests that the observed acceleration upon adding 1,5-cod should not stem from a change in rheology. We also confirmed that dramatic changes in the physical form of the reaction mixtures did not occur during the reaction (Fig. 9b). We also noted that the conversion rate of the reactions that contained cyclooctane or that were free of additives significantly decreased at ~30 min. These results are consistent with the TEM analysis, which revealed the formation of higher aggregates of dense palladium particles after grinding for 30 min (Supplementary Figures 10 and 11).

## Discussion

We have developed a rational strategy for a potentially general and scalable solid-state palladium-catalyzed cross-coupling reaction using mechanochemistry. Whereas the palladium-catalyzed cross-coupling of neat liquids proceeds readily in ball mills, similar reactions using solid reactants remain challenging. However, we discovered that the addition of small amounts of olefins dramatically accelerates the C–N cross-coupling of such solid substrates. The examination of palladium nanoparticles, which were obtained from these reaction mixtures, by transmission electron microscopy (TEM) suggested that the olefin additives can act as efficient molecular dispersants for the palladium catalysts in solid-state media and thus facilitate this challenging

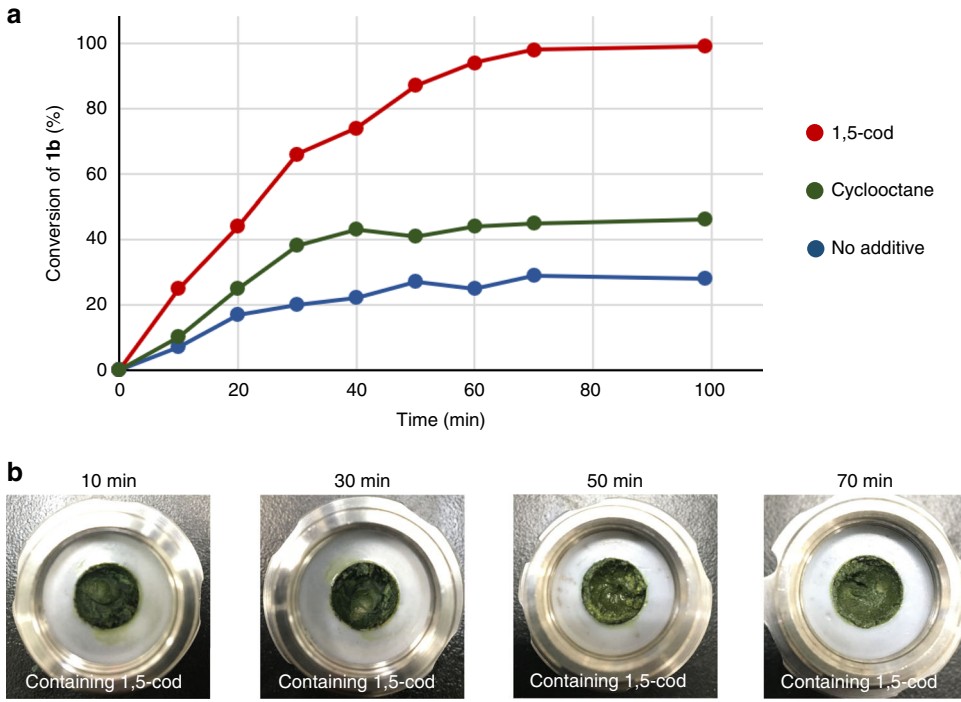

**Fig. 9** The kinetic study. **a** The kinetics of the reaction in the presence of 1,5-cod were found to be relatively straightforward (modeled as simple first order). This result suggests that the observed acceleration effect should not stem from changes in the rheology. **b** Dramatical changes in the physical form of the reaction mixtures containing 1,5-cod were not observed as the reaction progressed

solid-state cross-coupling. We anticipate that the strategy developed in this study could unlock broad areas of chemical space for palladium-catalyzed solid-state syntheses of valuable synthetic targets in various scientific fields.

## Methods

**Representative procedure for the solid-state coupling**. Aryl halide **1** (0.5 mmol), diarylamine **2** (0.5 mmol), and Pd(OAc)$_2$ (0.025 mmol) were placed in a ball-milling jar (1.5 mL) that contained a grinding ball (stainless steel; diameter: 0.3 cm). After the jar had been placed in a glovebox, $t$-Bu$_3$P (0.025 mmol) and Na (O-$t$-Bu) (0.75 mmol) were added. The jar was then removed from the glovebox, and 1,5-cod (0.20 μL mg$^{-1}$) was added in air. The jar was then placed in the ball mill, and after grinding (30 Hz, 99 min), the reaction mixture was passed through a short silica gel column (eluent: EtOAc). The crude product was subsequently purified by flash column chromatography (SiO$_2$; CH$_2$Cl$_2$/hexane, typically 0:100→15:90) to give the corresponding amination product **3**.

## Data availability

For full characterization data including NMR spectra of new compounds and experimental details, see the Supplemental Information. All relevant data underlying the results of this study are available from the corresponding authors upon reasonable request.

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

## Acknowledgements

This work was supported by MEXT/JSPS KAKENHI: Grant Numbers JP17H06370 and JP18H03907 and Institute for Chemical Reaction Design and Discovery (ICReDD) which was established by World Premier International Research Initiative (WPI), MEXT, Japan.

## Author contributions

K.K. and H.I. conceived and designed the study and co-wrote the paper. K.K. and S.T. performed the chemical experiments and analyzed the data. K.K. and Y.H. performed the transmission electron microscopy analysis. All authors discussed the results and commented on the manuscript.

## Additional information

**Competing interests:** The authors declare no competing interests.

