## [Peer Review File · Nature Communications]

Reviewer #1 (Remarks to the Author):

The authors present a method to enable solid state C-N coupling reactions between diphenyl amines and aryl halides. This method relies on the addition of alkenes as Pd-dispersants to the reaction, which increases efficiency by stabilization of nanoparticles. Overall, the authors present a reasonable substrate scope and show the method can be scaled up to gram scale. Due to rationalization of the results, the addition of an additive to a solid state reaction that can stabilize active nanoparticles, this could lead to similar developments for other reactions that are tricky in solution due to solubility issues or where the elimination of a reaction solvent is required. Therefore, this manuscript is definitely of interest for the readership of Nat. Commun.

Conceptually there are several topics lacking in the discussion regarding this approach. Firstly, solvent waste is indeed an issue but solvent recycling which is a very effective way to circumvent this is not mentioned in the introduction. Secondly, solvent waste is maybe not even most pronounced at the stage of the reaction but more so for the work-up and purification. Actually, although the solvent is eliminated in these reactions the first step after completion is dissolving the mixture in EtOAc and filtering it through celite. The latter is beyond the scope of this research but definitely a future challenge that should be addressed.

Furthermore a few details should be addressed to better support the conclusion of the manuscript:

- The sole support, although strong, for the hypothesis is a TEM image of one reaction with an olefin. It would be worthwhile to show how general this is. Related to this: Do these particles grow in time? Some distributions at different reaction times would be valuable. Also, as one hypothesis early in the text mentions that a reason for improved reactions in liquid over solid might be fast deactivation by aggregation in solid state reactions. So demonstrating the timescale for the aggregation is essential to the claims in the manuscript. Also, what happens with other Pd precursors?

- The colour observation is mentioned in the text and nicely shown in a figure. However, these are not discussed. Was it a general observation that reaction that worked well were green waxy solids and those that did not were light-yellow? and why waxy? The formation of disperse nanoparticles can hardly be the reason for that.

- One slight worry is the limited substrate scope on the amine side. How do the authors rationalize the reactivity trend between reactions to yield 3b, c and d. There is also a trend here compared to the melting points and although the authors show nice thermography, localized temperatures upon impact could be higher. The explanation for the lack of reactivity for carbazole seems odd as carbazole can be readily coupled in solution so why would strong coordination be any different there. Please clarify the "extremely-high-concentration conditions" (ratios are typical of solution phase reactions). This really leads to the question if the authors tried any other solid amines? Negative results apart from carbazole can be useful for the readers to understand the limitations!

Minor: Some citations seem to be wrong (e.g. 17 the page numbers and year are wrong. Should be: *Angew Chem Int Ed Engl.* 2014 Aug 25;53(35):9321). The authors should carefully check these.

Reviewer #2 (Remarks to the Author):

The manuscript by Ito and co-workers describe a solid state palladium catalyzed C-N coupling reaction for the synthesis of various substrate including molecules of importance to the solar cell community. Although the work performed sheds some light on a possible olefin complexation that allows for increased reactivity of palladium nanoparticles. The work presented is consistent with the hypothesis the authors reported however it is also potentially consistent with work performed by Stuart James on the effect rheology has on mechanochemical reactions. In this report, (B. P. Hutchings, D. E. Crawford, L. Gao, P. Hu, S. L. James, *Angew Chem Int Ed Engl* 2017, 56, 15252-15256.) James discovered a feedback kinetics based on various additives to the reaction, which in turn changes the rheology of the materials which changes the rates of the reaction. Similar to what was observed in this report, the materials look very different when milled under different conditions. The authors need to address whether the rate increase is from the olefin or the change

in rheology. In the optimization studies the authors should explain why they believe 60 microliters is optimum given their detailed hypothesis. In addition, some reactions were conducted for 99 minutes while others at 60 minutes, this makes it nearly impossible to determine the importance of the amount of olefin used and how any potential rheology changes would affect yield. The authors also demonstrated to be able to conduct the reaction on gram scale, suggesting this methodology would be applicable in industry. Various reports (D. Tan, T. Friščić, *Eur. J. Org. Chem.* 2018, 18-33; b) D. Crawford, J. Casaban, R. Haydon, N. Giri, T. McNally, S. L. James, *Chem Sci* 2015, 6, 1645-1649; c) D. E. Crawford, L. A. Wright, S. L. James, A. P. Abbott, *Chem Commun* 2016, 52, 4215-4218.) have shown the ability for mechanochemical reactions to be conducted in kilogram scale, which should be the expected norm if industry applicability is being claimed

Reviewer #3 (Remarks to the Author):

In this article Kubota, Ito and co-workers describe a solid-state cross-coupling reaction between amines and different aryl bromides. The authors employ a ball mill setup, and report here that the use of olefin additives is essential for a high yield. The role of the olefin is to act as a molecular dispersant, avoiding agglomeration of palladium into inactive metal clusters.

The main achievement of this article is that this method enables to obtain high yields for those substrates with very low solubility in the typical organic solvents used for this C-N coupling. As the authors highlight, this is a good method for coupling polyaromatic halides, and indeed this methodology affords the products in high yields and it can be scaled up to a few mmols.

One of the author's claims is the development of solid-state cross-coupling reactions. In general, solid-state catalysis is a very underdeveloped area. For catalysis to occur in the solid state, the molecules (reagents) must be able to enter the extended structure of the catalyst. I do not think that the authors have proved that this is solid catalysis in this sense. The authors use a mixture of Pd complex and a ligand. The structure of the catalyst is not well defined. In addition, an obvious advantage of solid-catalysis would be to obtain additional selectivity due to the catalysis happening in a constrained solid network, and that differs from the results obtained in solution catalysis.

In summary, despite this important advance in the methodology to synthesize polyaromatic amines in higher yield, it is my opinion that this paper is not of exceptional novelty for *NCOMMS*, and I therefore recommend publication in a more specialized catalysis journal.

Reviewers' comments:

Reviewer #1 (Remarks to the Author):

The authors present a method to enable solid state C-N coupling reactions between diphenyl amines and aryl halides. This method relies on the addition of alkenes as Pd-dispersants to the reaction, which increases efficiency by stabilization of nanoparticles. Overall, the authors present a reasonable substrate scope and show the method can be scaled up to gram scale. Due to rationalization of the results, the addition of an additive to a solid state reaction that can stabilize active nanoparticles, this could lead to similar developments for other reactions that are tricky in solution due to solubility issues or where the elimination of a reaction solvent is required. Therefore, this manuscript is definitely of interest for the readership of Nat. Commun.

Conceptually there are several topics lacking in the discussion regarding this approach.

Comment 1-1:

Firstly, solvent waste is indeed an issue but solvent recycling which is a very effective way to circumvent this is not mentioned in the introduction.

Response 1-1: We thank the reviewer for pointing out the lack of discussion on solvent recycling. We have added in the following sentence in the introduction of the revised manuscript.

“.....as organic solvents usually account for approximately 80–90% of the total mass used in any organic reaction.¹⁻³ Although solvent recycling is a very effective way to reduce solvent waste, organic chemists should focus on (re)designing organic syntheses to use less or no solvent. In this context, solid-state organic transformations.....”

Comment 1-2:

Secondly, solvent waste is maybe not even most pronounced at the stage of the reaction but more so for the work-up and purification. Actually, although the solvent is eliminated in these reactions the first step after completion is dissolving the mixture in EtOAc and filtering it through celite. The latter is beyond the scope of this research but definitely a future challenge that should be addressed.

Response 1-2: We agree with this reviewer's opinion. Our future work will focus on how to minimize the use of organic solvents during work up and purification.

Furthermore a few details should be addressed to better support the conclusion of the manuscript:

Comment 1-3:

- The sole support, although strong, for the hypothesis is a TEM image of one reaction with an olefin. It would be worthwhile to show how general this is. Related to this: Do these particles grow in time? Some distributions at different reaction times would be valuable. Also, as one hypothesis early in the text mentions that a reason for improved reactions in liquid over solid might be fast deactivation by aggregation in solid state reactions. So demonstrating the timescale for the aggregation is essential to the claims in the manuscript.

Response 1-3: The palladium nanoparticles generated *in situ* in the crude reaction mixtures that contain cyclooctane or that are free of additives have been characterized at different time points. The TEM images of the palladium particles generated upon grinding for 10 min did not show higher aggregates (Figures S5 and S6). On the other hand, the images of the palladium species generated upon grinding for 30 min showed dense aggregated particles (Figures S5 and S6). These results are consistent with a significantly slower conversion of the starting material (~ 30 min) in both reactions (cyclooctane or no additive). The kinetics data is shown below and the details are discussed in Response 2-1. The TEM images have been added to the Supporting Information as Figures S5 and S6.

Figure S5. Aggregation of palladium particles as a function of time in the reaction mixture containing cyclooctane. Scale bars in the TEM images (bottom left) indicate 20 nm.

Figure S6. Aggregation of palladium particles as a function of time in the reaction mixture without additives. Scale bars in the TEM images (bottom left) indicate 20 nm.

Figure 5. The conversion of the solid-state C–N cross-coupling reaction between **1b** and **2a** as a function of the reaction time in the presence of 1,5-cod or cyclooctane, and in the absence of additives.

Comment 1-4:

Also, what happens with other Pd precursors?

Response 1-4: The solid-state coupling reaction using Pd(dba)₂ instead of Pd(OAc)₂ in the absence of 1,5-cod was sluggish (19% NMR yield). In contrast, the reaction in the presence of 1,5-cod proceeded smoothly to give the coupling product in high yield (93% NMR yield). These results suggest that our strategy can be applied to other Pd precursors. We have added these results to Scheme S1 in the Supporting Information.

Scheme S1. The use of Pd(dba)₂ instead of Pd(OAc)₂.

Comment 1-5:

- The colours observation is mentioned in the text and nicely shown in a figure. However, these are not discussed. Was it a general observations that reaction that worked well were green waxy solids and those that did not were light-yellow? and why waxy? The formation of disperse nanoparticles can hardly be the reason for that.

Response 1-5: It was not a general observation, given that the color of the crude reaction mixtures containing 1,5-cod were almost identical those containing cyclooctane. We assume that the waxy solids were formed due to the small amounts of liquids that were used in these reactions.

Comment 1-6:

- One slight worry is the limited substrate scope on the amine side. How do the authors rationalize the reactivity trend between reactions to yield 3b, c and d. There is also a trend here compared to the melting points and although the authors show nice thermography, localized temperatures upon impact could be higher. The explanation for the lack of reactivity for carbazole seems odd as carbazole can be readily coupled in solution so why would strong coordination be any different there. Please clarify the "extremely-high-concentration conditions" (ratios are typical of solution phase reactions). This really leads to the question if the authors tried any other solid amines? Negative results apart from carbazole can be useful for the readers to understand the limitations!

Response 1-6: The reasons for the reactivity difference of the substrates are most likely complex and may be due to a combination of several factors such as changes in the chemical composition and the particle size of the reactants. The hypothesis that the melting points could determine the reactivity might be an oversimplification, considering that 2,2'-dinaphthylamine (**2e**) (mp. 172-176 °C), which has a much higher melting point than

bis(4-methoxyphenyl)amine (**2d**) (mp. 100-104 °C), affords a better yield than **2d** in this reaction.

Since the solvent-free solid-state reaction can be regarded as the reaction under extremely-high-concentration conditions, the reagents and catalysts in the solid state interact with each other much more strongly than in solution. This could be an explanation for the observed low reactivity of carbazole (**2g**), which may strongly coordinate to any off-cycle palladium species, tantamount to catalyst deactivation. We have already shown all amines tested in this study in the original manuscript, i.e., we do not have any further negative results to add.

To clarify "the extremely-high-concentration conditions", we modified the following sentence.

Original sentence:

Conversely, the reaction did not proceed for carbazole (**2g**), which is probably due to the strong coordination of **2g** to the active Pd(0) species under the extremely-high-concentration conditions that may deactivate the catalyst.

The revised sentence:

Conversely, the reaction did not proceed for carbazole (**2g**). *As solvent-free solid-state reactions can be regarded as reactions that proceed under extremely-high-concentration conditions, the reagents and catalysts in the solid state interact much more strongly with each other than those in solution. This could tentatively explain the observed low reactivity of **2g**, which could potentially coordinate strongly to any off-cycle palladium species, which would lead to catalyst deactivation.*

Comment 1-7:

Minor: Some citations seem to be wrong (e.g. 17 the pagenumbers and year are wrong. Should be: Angew Chem Int Ed Engl. 2014 Aug 25;53(35):9321). The authors should carefully check these.

Response 1-7: This error has been corrected.

Reviewer #2 (Remarks to the Author):

The manuscript by Ito and co-workers describe a solid state palladium catalyzed C-N coupling reaction for the synthesis of various substrate including molecules of importance to the solar cell community. Although the work performed sheds some light on a possible olefin complexation that allows for increased reactivity of palladium nanoparticles.

Comment 2-1:

The work presented is consistent with the hypothesis the authors reported however it is also potentially consistent with work performed by Stuart James on the effect rheology has on mechanochemical reactions. In this report, (B. P. Hutchings, D. E. Crawford, L. Gao, P. Hu, S. L. James, *Angew Chem Int Ed Engl* 2017, 56, 15252-15256.) James discovered a feedback kinetics based on various additives to the reaction, which in turn changes the rheology of the materials which changes the rates of the reaction. Similar to what was observed in this report, the materials look very different when milled under different conditions. The authors need to address whether the rate increase is from the olefin or the change in rheology.

Response 2-1: Thank you very much for the insightful comment. As mentioned, James and co-workers have discovered unusual sigmoidal kinetics in the mechanochemical Knoevenagel condensation of vanillin and barbituric acid, in which the physical form of the reaction mixture dramatically changes from a free-flowing powder to a plastic-like appearance during the reaction. Based on the several control experiments, they concluded that this dramatical change in rheology results in a rapid increase in the reaction rate. To address whether the rate increase is due to the olefin or the change in rheology, the kinetics of the reactions under different conditions (1. In the presence of 1,5-cod, 2. In the presence of cyclooctane, 3. No additives) were measured (Figure 5a). As periodic sampling of the reaction runs requires stopping the mill and opening the jar, each data point was obtained from an individual reaction. The kinetics of the reaction between **1b** and **2a** in the presence of 1,5-cod were found to be relatively straightforward, i.e., they could be satisfactorily modelled by simple first-order kinetics. This result suggests that the observed acceleration upon adding 1,5-cod should not stem from a change in rheology. We also confirmed that dramatic changes in the physical form of the reaction mixtures did not occur during the reaction (Figure 5b). We also noted that the conversion rate of the reactions that contained cyclooctane or that were free of additives significantly decreased at ~30 min. These results are consistent with the TEM analysis, which revealed the formation of higher aggregates of

dense palladium particles after grinding for 30 min. The kinetics data and pictures of the crude mixture containing 1,5-cod have been added as Figure 5 in the revised manuscript to discard the possibility that changes of the rheology could affect the olefin-accelerated solid-state coupling reaction.

Figure 5. The conversion of the solid-state C–N cross-coupling reaction between 1b and 2a as a function of the reaction time in the presence of 1,5-cod or cyclooctane, and in the absence of additives. a, The kinetics of the reaction in the presence of 1,5-cod were found to be relatively straightforward (modelled as simple first order). This result suggests that the observed acceleration effect should not stem from changes in the rheology. b, Dramatic changes in the physical form of the reaction mixtures containing 1,5-cod were not observed as the reaction progressed.

Comment 2-2:

In the optimization studies the authors should explain why they believe 60 microliters is optimum given their detailed hypothesis. In addition, some reactions were conducted for 99 minutes while others at 60 minutes, this makes it nearly impossible to determine the importance of the amount of olefin used and how any potential rheology changes would affect yield.

Response 2-2: We have already included a reasonable optimization study in Table S1 in the Supporting Information. The results show that the reaction using 60 μL of 1,5-cod provides the product in quantitative yield (99%), while the use of 40 μL of 1,5-cod affords the product in 90% yield. The use of 80 μL of 1,5-cod or more decreased the product yield. The reaction time for all solid-state reactions described in the manuscript is 99 min. The reviewer may have misread the manuscript.

Comment 2-3:

The authors also demonstrated to be able to conduct the reaction on gram scale, suggesting this methodology would be applicable in industry. Various reports (D. Tan, T. Friščić, Eur. J. Org. Chem. 2018, 18-33; b) D. Crawford, J. Casaban, R. Haydon, N. Giri, T. McNally, S. L. James, Chem Sci 2015, 6, 1645-1649; c) D. E. Crawford, L. A. Wright, S. L. James, A. P. Abbott, Chem Commun 2016, 52, 4215-4218.) have shown the ability for mechanochemical reactions to be conducted in kilogram scale, which should be the expected norm if industry applicability is being claimed

Response 2-3: We have removed the word “industrial applications” on page 13. However, we believe that this demonstration represents an important milestone toward future industrial applications.

Original sentence:

This result clearly demonstrates the potential utility of the present solvent-free protocol for large-scale preparations and industrial applications.

Revised sentence:

This result clearly demonstrates the potential utility of the present solvent-free protocol for large-scale preparations.

Reviewer #3 (Remarks to the Author):

In this article Kubota, Ito and co-workers describe a solid-state cross-coupling reaction between amines and different aryl bromides. The authors employ a ball mill setup, and report here that the use of olefin additives is essential for a high yield. The role of the olefin is to act as a molecular dispersant, avoiding agglomeration of palladium into inactive metal clusters. The main achievement of this article is that this method enables to obtain high yields for those substrates with very low solubility in the typical organic solvents used for this C-N coupling. As the authors highlight, this is a good method for coupling polyaromatic halides, and indeed this methodology affords the products in high yields and it can be scaled up to a few a mmols.

Comment 3-1:

One of the author's claims is the development of solid-state cross-coupling reactions. In general, solid-state catalysis is a very underdeveloped area. For catalysis to occur in the solid state, the molecules (reagents) must be able to enter extended structure of the catalyst.

I do not think that the authors have proved that this is solid catalysis in this sense. The authors use a mixture of Pd complex and a ligand. The structure of the catalyst is not well defined.

Response 3-1: For a solution study, Hartwig and co-workers have reported that the palladium-catalyzed C-N cross-coupling using $t\text{Bu}_3\text{P}$ proceeds via the formation of three-coordinate oxidative-addition complexes, which suggests that $[(t\text{-Bu}_3\text{P})\text{Pd}(0)]$ is the active species of the reaction (*J. Am. Chem. Soc.* **2004**, *126*, 5344.). We haven't examined whether the active species in our solid-state reaction is $[(t\text{-Bu}_3\text{P})\text{Pd}(0)]$, but it seems highly unlikely given that $[(t\text{-Bu}_3\text{P})\text{Pd}(0)]$ is extremely unstable.

We do not fully understand the intended meaning of the comment "*the molecules (reagents) must be able to enter extended structure of the catalyst.*", but the coupling reaction requires the contact between Pd catalyst and reagents. Based on the PXRD analysis (Figure 4) in the manuscript, we have confirmed a solid-to-solid conversion without melting in the reaction. The Pd catalyst and reagents must thus be in contact in the solid state. Thus, the present cross-coupling reaction can be regarded as a solid-state cross-coupling process.

Comment 3-2:

In addition, an obvious advantage of solid-catalysis would be to obtain additional selectivity due to the catalysis happening in a constrained solid network, and that differs from the results obtained in solution catalysis.

Response 3-2: This manuscript focuses on (re)designing palladium-catalyzed cross-coupling reactions for solid-state media. The synthetic utility of our solid-state approach has been demonstrated by the rapid synthesis of the hole-transporting material shown in Scheme 1b. Our solid-state reaction affords higher yields than under conventional solution conditions. This demonstrates a strong advantage of the solid catalysis.

We agree that the discovery of unique selectivity in the solid-state would be another interesting topic; however, this is not our main concern in the present study. The intermolecular interactions are generally strong and anisotropic in comparison to those in solution. The current solid-state reaction does not strongly rely on this feature. We are currently investigating solid-state reactions that exhibit selectivity that cannot be achieved in solution.

Comment 3-3:

In summary, despite this important advance in the methodology to synthesize polyaromatic amines in higher yield, it is my opinion that this paper is not of exceptional novelty for NCOMMS, and I therefore recommend publication in a more specialized catalysis journal.

Response 3-3: The reviewer may miss the main achievement of this work. The present paper does not offer an incremental improvement for the synthesis of triaryl amines, but the discovery of molecular dispersants for palladium-based catalysts in solid-state media, which should allow realizing a variety of solvent-free solid-state palladium-catalyzed cross-coupling reactions. More importantly, we expect the effect of olefins discovered here to be a robust and general strategy for the stabilization of palladium-based catalysts in the solid state, which may have significant impact on a broad range of scientific fields such as organic synthesis, materials science, solid-state chemistry, and green chemistry. Beyond the academic interests, the present method could inspire the development of efficient solvent-free protocols for large-scale preparations and industrial applications. Thus, we believe that the quality, novelty, and significance of the results described in this manuscript should attract substantial interest from broad segments of the readership of *Nature Communications*.

Other Non-Scientific Changes:

1. Information on the affiliation has been updated.
2. The acknowledgements in the revised manuscript have been updated.

Reviewer #1 (Remarks to the Author):

The author present a a solid state palladium catalyzed C-N coupling reaction that allows for a rapid synthesis of a variety of interesting molecules. The revision have led to significant improvements and I recommend publication in Nature Communications. In particular, I applaud the addition of the "kinetic study" section which helps support the claims made and provides much better insight into what is going on.

Reviewer #2 (Remarks to the Author):

After addressing the comments the article is now ready to be published

Reviewers' comments:

Reviewer #1 (Remarks to the Author):

The author present a solid state palladium catalyzed C-N coupling reaction that allows for a rapid synthesis of a variety of interesting molecules. The revision have led to significant improvements and I recommend publication in Nature Communications. In particular, I applaud the addition of the "kinetic study" section which helps support the claims made and provides much better insight into what is going on.

Reviewer #2 (Remarks to the Author):

After addressing the comments the article is now ready to be published.

Reviewer #3 (Remarks to the Author):

Comment 3-1:

It is my opinion that the article lacks insight into how the catalysis in the solid-state proceeds. I am however aware that such a study would be extremely challenging, as for example the set-up is unlike to be compatible with an in-situ EXAFS cell, for example. The authors might mention these challenges in the modified manuscript.

Response 3-1:

We added the following sentence describing the suggested challenges in ref. 61.

Although further investigations on how the catalyst diffuses in the solid-state media should provide a deep understanding the reaction mechanism, such a study would be extremely challenging, i.e., the mechanochemical reaction set-up is unlike to be compatible with an in-situ extended X-ray absorption fine structure (EXAFS) analysis.